# Effects of Rehabilitation Exercise on Cardiovascular Risk Factors and Muscle Cross-Sectional Area in Overweight Patients with Low Back Pain

**DOI:** 10.3390/healthcare9070809

**Published:** 2021-06-27

**Authors:** Won-Moon Kim, Su-Ah Lee, Yun-Jin Park, Yong-Gon Seo

**Affiliations:** 1Department of Sports Science, Dongguk University, 123 Dongdae-ro, Gyeongju-si 38066, Gyeongsangbuk-do, Korea; kimwonmoon3426@hanmail.net; 2Department of Sports Science, Hanyang University, 55 Hanyang Daehak-ro, Sangnok-gu, Ansan-si 15588, Gyeonggi-do, Korea; lko7774@naver.com; 3Department of Health Rehabilitation, Osan University, 45 Cheonghak-ro, Osan-si 18119, Gyeonngi-do, Korea; africca3535@gmail.com; 4Samsung Medical Center, Department of Orthopedic Surgery, Division of Sports Medicine, 81 Irwon-ro, Gangnam-gu, Seoul 06351, Korea

**Keywords:** low back pain, obesity, disability evaluation, exercise therapy

## Abstract

Limited studies exist on the effects of exercise therapy on obese and normal-weight patients. Herein, we investigated the effect of a 12-week rehabilitation exercise program on cardiovascular risk factors, Oswestry Disability Index (ODI), and change in the cross-sectional area (CSA) of lumbar muscles in patients with obesity and normal-weight low back pain (LBP). LBP patients were allocated to the overweight LBP group (OLG; *n* = 15) and normal-weight LBP group (NLG; *n* = 15). They performed a rehabilitation exercise program three times per week for 12 weeks. Cardiovascular risk factors, ODI score, and lumbar muscle CSA were assessed pre- and post-intervention. Body composition, body weight (*p* < 0.001), and body mass index (*p* < 0.001) significantly improved after the exercise intervention in OLG. Body fat percentage significantly decreased in both groups, but OLG (*p* < 0.001) showed slightly greater improvement than NLG (*p* = 0.034). Total cholesterol (*p* = 0.013) and low-density lipoprotein (*p* = 0.002) significantly improved in OLG. ODI score improved significantly in both groups (*p* = 0.000). Lumbar muscle CSA showed a significant difference in the context of the time result (*p* = 0.008). OLG showed a significant improvement post-intervention (*p* = 0.003). The rehabilitation exercise program was more beneficial on cardiovascular risk factors and change in lumbar muscle CSA in OLG, suggesting an intensive exercise intervention needed for overweight patients with LBP.

## 1. Introduction

Obesity refers to the over-accumulation of fat tissues in the body and is associated with an imbalance between energy intake and consumption, inappropriate diet, lack of physical activities, mental stress, and endocrine disorder [1]. An increasing musculoskeletal disorder with low back pain (LBP) has been reported in obese middle-aged women [2,3]. Additionally, decreased physical activity and irregular dietary habits can cause increasing abdominal obesity, cardiovascular disease (CVD), and metabolic disorders, resulting in chronic LBP [4,5].

Chronic LBP contributes to physical inactivity and decreased lumbar muscle strength and can cause muscle imbalance and decreased ability to perform activities of daily living and functional activities [6]. In general, the intervention includes medication and injection for the treatment of LBP, to reduce pain and improve lumbar function. Exercise therapy is one of the common interventions for the treatment of LBP and is reported to be a more effective intervention for chronic LBP [7].

Of patients with chronic LBP, more than 80% showed decreased lumbar muscle strength [8]. Increased body fat and decreased lean body mass (LBM) in obesity serve as risk factors for CVD and exacerbate chronic LBP [9]. Previous studies [10,11] have reported that exercise therapy is effective in reducing body fat and LBP and increasing muscle strength and flexibility in obese LBP patients.

LBP in obese patients is associated with the presence of cardiovascular risk factors, decreased ODI score, and decreased cross-sectional area (CSA) of lumbar muscles [1]. Several studies [12,13,14,15] have been conducted on the effectiveness of exercise therapy in patients with LBP, but no consensus was achieved between the studies. According to a review article [1], the highest adherence rate occurred with resistance exercise than other types included aerobic exercise and have reported the types may provide a greater overall impact in %BF in overweight-obese LBP patients.

Clinical exercise programs for LBP have been extensively investigated, but relatively few have focused on overweight individuals. Therefore, this study aimed to compare the effects of a 12-week rehabilitation exercise program consisted of various resistance exercises on cardiovascular risk factors, ODI score, and change in lumbar muscle CSA between overweight and normal-weight LBP patients.

## 2. Materials and Methods

### 2.1. Study Design

This study was conducted from 1 September 2020 to 30 April 2021. The G-power program (Heinrich Heine University Düsseldorf, Düsseldorf, Germany, version 3.1.9.4) [16] was used to determine the sample size in which the effect size = 0.25, α = 0.05, power = 0.80%. According to the criteria, the minimal sample size was calculated as 34. However, thirty female patients with LBP with moderate disability (ODI = 21–40%) participated in this study. They were allocated to two groups according to the Clinical Practice Guidelines for Overweight and Obesity in World Health Organization [17]. Patients whose body mass index (BMI) was >25 kg/m^2^ were assigned to the overweight group (OLG, *n* = 15) and the remaining in the normal-weight group (NLG, *n* = 15). The inclusion criteria were as follows: Patients with no CVD, no history of orthopedic or neurosurgery, and no musculoskeletal disorder other than LBP.

The study protocol was approved by the Institutional Review Board (IRB) of DongGuk University (IRB No.: DGU-20200022), and the study was conducted in compliance with the 1964 Helsinki Declaration. Informed consent was obtained from each patient before participation in the present study.

### 2.2. Outcome Assessments

#### 2.2.1. Body Composition

For body composition assessment, the participants were restricted from eating and performing excessive physical activities for 12 h before the measurements. Body composition using a piece of equipment (Inbody 270, Biospace, Seoul, Korea) based on bioelectrical impedance analysis was used for evaluating the body weight (BW), LBM, BMI, and body fat percentage (%BF).

#### 2.2.2. Blood Profile

For hematological analysis, the participants were instructed to fast for 12 h, and blood collection was obtained after resting for 30 min. Blood sampling involved the collection of 5.0 mL of whole blood from the brachial artery at the antecubital fossa using a blood collection container (SST, Greiner Bio One, Frickenhausen, Germany) equipped with a vacuum needle. The collected blood samples were coagulated at room temperature and subsequently centrifuged (Centrifuge, PLD-01, Taiwan) for 10 min at 3000 rpm. From the separated supernatant, a portion was transferred to a serum separation tube and freeze-dried. The portion remaining in the blood collection container was refrigerated and subsequently analyzed using the Green Cross Corporation. CHOL2 (Roche, Grenzach-Wyhlen, Germany) was used as the reagent for the enzymatic colorimetric assay using a chemistry analyzer (Cobas 8000 c702, Roche, Mannheim, Germany). Low-density lipoprotein cholesterol (LDL-C) level was calculated using the formula [LDL-C=TC-HDL-C-(TG/5)] [18] based on measured values for total cholesterol (TC), triglycerides (TG), and high-density lipoproteins cholesterol (HDL-C). For Homeostatic Model Assessment for Insulin Resistance (HOMA-IR), blood glucose and insulin levels were measured. The unit of measurement for blood glucose was converted to mmol/l by dividing mg/dl by a conversion constant of 18, after which the values were analyzed using the HOMA-IR formula to derive the results [19]: Insulin* Glucose/22.5.

#### 2.2.3. Cross-Sectional Area

Computed tomography (Sytec-Sri, GE, Boston, MA, USA) was performed to measure the CSA of the lumbar muscles. The test was conducted by a single radiologist with more than 20 years of experience who performed all imaging and measurements. On the monitor screen of the picture archiving and communication system, the range of CSA of the erector spinae muscle was analyzed, and the CSA (cm^2^) was subsequently calculated (Figure 1).

#### 2.2.4. Oswestry Disability Index (ODI)

The ODI questionnaire is the most commonly used for evaluating disability in performing activities of daily living due to LBP. The questionnaire was designed as 10 items (pain intensity, personal care, lifting, walking, sitting, sleeping, standing, sex life, traveling, and social life) scored from 0 points defined as no discomfort to 5 as most discomfort. A lower total score means a lower physical disability [20].

### 2.3. Rehabilitation Exercise Programs

In the present study, a rehabilitation exercise program was equally applied to OLG and NLG for 12 weeks. The rehabilitation exercise program was modified by referring to previous studies [21,22] and consisted of stabilization and strengthening exercises for trunk muscles. The rehabilitation exercise program was categorized into two phases. The first phase focused on muscle activation to stimulate the trunk muscles through abdominal bracing and drawing-in method by breathing. The second phase is an advanced stage to improve muscle strengthening using plank positions by isolating the abdominal muscles with small tools to enhance the requirement of balance. The exercise intensity was set to the rating of perceived exertion of 13 (somewhat hard) to 15 (hard). This program was conducted three times per week for 12 weeks. The exercise time was 70 min and the resting time was set to 30 s after each set and 50 s between exercises (Table 1).

### 2.4. Statistical Analysis

A descriptive analysis was performed to determine the mean and standard deviation. An independent sample *t*-test was performed to confirm the homogeneity of baseline variables between the groups, and the normality test was performed using the Kolmogorov–Smirnov test. Two-way repeated analysis of variance was performed to compare the effect of the rehabilitation exercise program between the groups. Paired *t*-tests were performed to analyze the change in the pre- to post-intervention in each group if there was a significant difference in the interaction effect. Data analysis was performed using SPSS version 22.0 Windows (IBM Corp, Armonk, NY, USA), and the significance of all data was set to *p* < 0.05.

## 3. Results

Thirty patients with LBP participated in this study, but one participant dropped out due to personal reasons in OLG. There were no differences in the baseline data except in BMI between the two groups before the study (Table 2).

Regarding body composition, changes in BW showed a significant difference in the time (F = 21.921, *p* < 0.001) and interaction between the group and time (F = 5.607, *p* = 0.025), but there was no difference between the two groups (F = 0.798, *p* = 0.380). Paired *t*-test results showed a significant change after intervention in OLG (*p* < 0.001). There was a significant difference in BMI between the groups (F = 23.628, *p* < 0.001) and %BF was also found between the two groups (F = 47.749, *p* < 0.001).

In terms of cardiovascular risk factors, changes in TC revealed a significant difference in the time (F = 6.272, *p* = 0.019) and interaction between the group and time (F = 5.148, *p* = 0.031), and post-hoc analysis showed more significant improvement in OLG (*p* = 0.013). Changes in TG showed a significant differences according to the time (F = 12.334, *p* = 0.002) but no significant difference according to interaction between the group and time (F = 2.002, *p* = 0.169) or between the two groups (F = 0.009, *p* = 0.926). There was no significant difference in HDL between the two groups (F = 0.098, *p* = 0.757). Changes in LDL showed a significant difference in the time (F = 7.614, *p* = 0.010) and interaction between the group and time (F = 7.769, *p* = 0.010). However, post-hoc analysis revealed more significant improvement in OLG (*p* = 0.002). Changes in HOMA-IR showed significant differences in the time only (F = 4.946, *p* = 0.035) (Figure 2).

In terms of the CSA of the lumbar muscles, the erector spinae muscle showed a significant differences in the time (F = 8.318, *p* = 0.008), and paired t-test for post-hoc analysis revealed significant improvement in OLG (*p* = 0.003). In terms of lumbar function, changes in ODI showed significant differences in the time (F = 127.151, *p* < 0.001) and interaction between the group and time (F = 20.370, *p* < 0.001). According to the result of post-hoc analysis, the improvement was seen after the intervention in both OLG (*p* = 0.003) and NLG (*p* < 0.001) (Figure 3). Table 3 shows the change of each variable before and after exercise intervention.

## 4. Discussion

This study demonstrated that a rehabilitation exercise program is a more beneficial intervention to improve cardiovascular risk factors (TC, TG and LDL) and CSA of lumbar spinal muscles in OLG than in NLG.

A decrease in physical activity is associated with a relatively higher risk of CVD related to increased body fat and decreased LBM [6]. Imbalance of body composition could increase the risk of musculoskeletal disorders included LBP and arthritis [23,24]. LBP causes weakness of the erector spinae muscle and physical inactivity, resulting in decreased muscle mass and cardiovascular functions [25]. Obesity is associated with the incidence of CVD and metabolic disorders and also can cause recurrent LBP [26,27]. Regular long-term exercise has been reported to improve blood lipid levels, including TC, TG, and LDL [28]. Lee et al. [25] have reported that lumbar muscle strengthening exercise for LBP patients with abdominal obesity significantly improved cardiovascular risk factors, including waist circumference (WC), visceral fat, and subcutaneous fat. These results were consistent with the findings of the present study. Changes in TC, TG, and LDL found in this study showed more significant improvement in OLG, and the finding was consistent with the results reported by Choi [29], which suggests that abdominal obesity increases the risk of LBP in women aged ≥50 years.

Patients with chronic LBP have decreased CSA and strength in the lumbar muscles [30]. In obese LBP patients, the erector spinae muscle of the lumbar region was in atrophy and fat infiltration [12]. Several studies have reported a significant change in the CSA of lumbar muscles in LBP patients after exercise intervention [9,31]. In the present study, there was a significant difference in the period between the groups, and OLG showed an increase in CSA after exercise intervention. A previous study [9] found that patients with abdominal obesity showed a significantly higher increase in lumbar extensor muscle strength after applying rehabilitation exercise of eight weeks compared with LBP patients with normal-weight individuals. Considering the relationship between the increase of lumbar muscle strength and increase of lumbar muscle CSA [32], the finding was consistent with the result of the present study, although this study did not measure the lumbar muscle strength. In young adult women with LBP, there was a positive association between fat infiltration and CSA in lumbar erector muscles [33]. In this study, an increase of CSA in obese LBP patients may be associated with an increase in lean mass. The exercise type applied in this study was stabilization training combined with intensive strengthening, which was reported as the most appropriate method of restoring the size of the multifidus muscle [12]. However, further study is needed to evaluate the ratio between fat infiltration and muscle mass for identifying the effect of the exercise program. 

Obesity is associated with limited function of the trunk, and ODI is used to evaluate the severity of pain and function in patients with chronic LBP [4,20]. The high level of adiposity may impair agonist muscle activation, leading to the functional limitation of low strength relative to body mass [34]. França et al. [7] reported that a significant decrease in ODI score was found after trunk stabilization exercise intervention in patients with chronic LBP. The present study showed more a decrease in ODI score in OLG. These findings are consistent with those of a previous study [1], which reported that obese patients with LBP experienced an improvement in their low back function more effectively as a result of greater body metabolism through rehabilitation exercise. The results may be because OLG showed a more significant change in body composition and improved muscle strength, resulting in improved spinal movement and function [6]. According to a study conducted by Urquhart et al. [35], the severity of LBP is associated with the amount of body fat. Thus, the more body fat, the greater the severity of LBP and the greater the functional inability caused by LBP. Considering these points, the authors believe that the reduction of body fat through rehabilitation exercise contributes to improved lumbar function. 

This study had some limitations. First, all participants were female patients aged 40–49 years. Women have a higher prevalence of LBP than men because the lumbar and paraspinal muscles in women are relatively weaker than those in men [36]. Therefore, it is difficult to generalize this result to all LBP patients in all age and sex groups. Further study is needed to compare the difference between the sexes. Second, the study groups were classified based on BMI and WC, and thus, the actual influence of subcutaneous and visceral fat was not considered. The effect of these factors should be considered for evaluating the relationship with LBP. Third, the sample size in this study was small. Future studies are needed to identify the findings of the present study with a large study population.

## 5. Conclusions

The rehabilitation exercise program for overweight patients with LBP can contribute to more improvement of cardiovascular risk factors (TC, TG, and LDL), lumbar function with ODI, increase of CSA of erector spinae muscles in the lumbar region. Therefore, overweight LBP patients should enroll in an intensive rehabilitation exercise program within a certain period to reduce the risk of cardiovascular disease and to improve lumbar functions.

## Figures and Tables

**Figure 1 healthcare-09-00809-f001:**
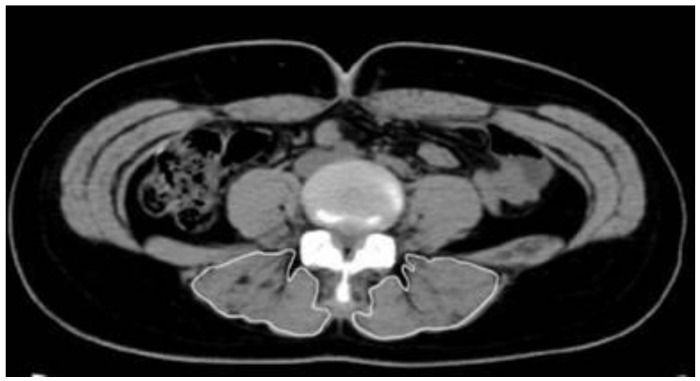
Measurement of the cross-sectional area of the erector spinae muscles.

**Figure 2 healthcare-09-00809-f002:**
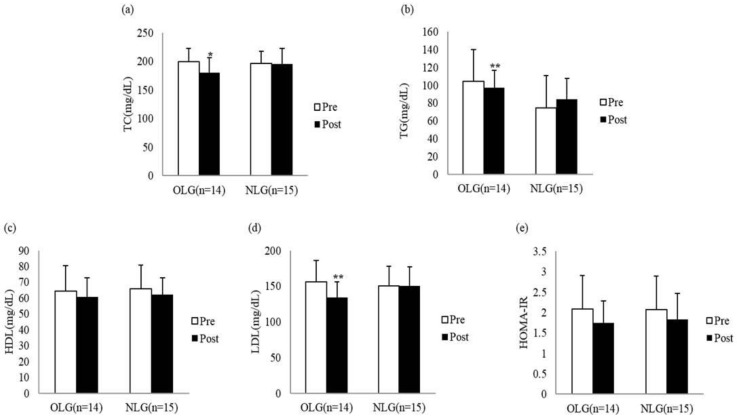
Pre- to post-intervention changes in cardiovascular risk factors between the two groups. (**a**) Total cholesterol, (**b**) triglycerides, (**c**) high-density lipoproteins, (**d**) low-density lipoproteins, and (**e**) homeostatic model assessment-insulin resistance. OLG, overweight low back pain group; NLG, normal weight low back pain group. *p* < 0.05 * and *p* < 0.01 ** are statistically significant.

**Figure 3 healthcare-09-00809-f003:**
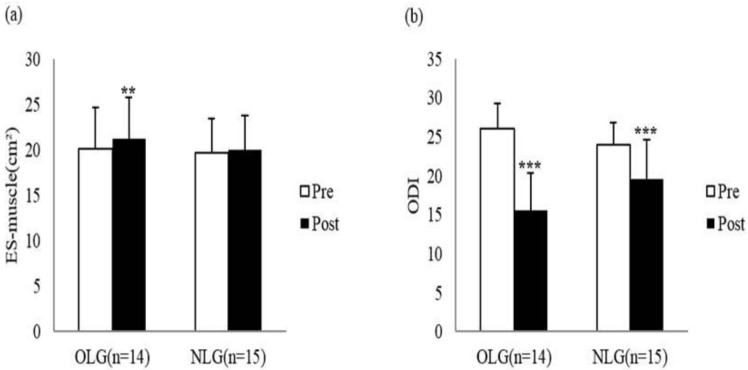
Pre- to post-intervention changes in the cross-sectional area of the erector spinae muscle and Oswestry disability index (ODI) between the two groups. (**a**) The cross-sectional area of the erector spinae muscle and (**b**) ODI. OLG, overweight low back pain group; NLG, normal-weight low back pain group. *p* < 0.01 ** and *p* < 0.001 *** are statistically significant.

**Table 1 healthcare-09-00809-t001:** The 12-week rehabilitation exercise program for this study.

Rehabilitation Exercise Program			
Exercise Types	Exercise Modes	Time	Intensity
Warm-up	Stretching for upper and lower body	5 min	RPE (10–13)
Main exercise (Phase 1)	Abdominal bracing	30 min	RPE (13–15) 20–30 reps 3 sets
	Bracing with bridge exercise in supine position		
	Bracing with side bridge exercise in side lying position		
	Bracing with bird dog exercise in quadruped position		
	Bracing with sit-up exercise in supine position		
Main exercise (Phase 2)	Front plank with knees extended	30 min	RPE (13–15) 20–30 reps 3 sets
	Side plank with knees extended		
	Upper/lower back extension exercise in prone position		
	Transverse plane core exercise in standing position		
	Step-up in standing position		
Cool-down	Stretching for upper and lower body	5 min	

RPE, rating of perceived exertion; reps, repetitions.

**Table 2 healthcare-09-00809-t002:** Characteristics of the study population.

Characteristics	OLG	NLG	*p*-Value
Numbers	14	15	–
Sex (female/male)	14/0	15/0	–
Age (years)	40.29 ± 2.89	40.13 ± 3.45	0.709
Height (cm)	158.18 ± 5.25	160.95 ± 5.08	0.160
Weight (kg)	64.13 ± 6.95	61.56 ± 4.23	0.235
ODI (score)	26.0 ± 3.26	24.0 ± 2.88	0.091
BMI (kg/m^2^)	26.62 ± 1.50	23.83 ± 0.81	0.000 ***

Data are presented as mean ± standard deviation. OLG, overweight low back pain group; NLG, normal-weight low back pain group; ODI, Oswestry disability index; BMI, body mass index. *p* < 0.001 *** is statistically significant.

**Table 3 healthcare-09-00809-t003:** Clinical outcomes after rehabilitation exercise program to each variable.

Variables	Group	Pre	Post	*t*-Value	*p*-Value
Body weight (kg)	OLG	64.13 ± 6.95	61.79 ± 6.30	5.232	0.000 ***
	NLG	61.27 ± 4.23	60.41 ± 3.82	1.578	0.137
Body mass index (kg/m^2^)	OLG	26.62 ± 1.50	25.12 ± 1.71	5.916	0.000 ***
	NLG	23.83 ± 0.81	23.46 ± 0.76	1.721	0.107
Body fat percentage (%)	OLG	35.72 ± 3.15	33.51 ± 3.00	6.101	0.000 ***
	NLG	27.41 ± 2.97	27.00 ± 2.91	2.357	0.034 *
Total cholesterol (mg/dL)	OLG	199.24 ± 23.22	180.29 ± 21.71	2.863	0.013 *
	NLG	199.07 ± 24.74	197.71 ± 27.17	0.205	0.840
Triglyceride (mg/dL)	OLG	104.64 ± 35.29	74.57 ± 19.97	3.829	0.002 **
	NLG	96.87 ± 36.32	84.07 ± 23.22	1.386	0.187
HDL (mg/dL)	OLG	64.50 ± 15.94	60.71 ± 11.96	0.906	0.381
	NLG	65.80 ± 15.13	62.20 ± 10.67	1.488	0.159
LDL (mg/dL)	OLG	155.64 ± 30.47	134.49 ± 21.92	3.794	0.002 **
	NLG	150.04 ± 27.79	150.15 ± 26.54	−0.020	0.984
HOMA-IR	OLG	2.09 ±0.82	1.74 ± 0.54	1.939	0.075
	NLG	2.07 ± 0.82	1.83 ± 0.64	1.249	0.232
Lumbar CSA (cm^2^)	OLG	20.16 ± 4.48	21.25 ± 4.74	−3.623	0.003 **
	NLG	19.71 ± 3.68	20.01 ± 3.77	−0.814	0.429
ODI (score)	OLG	26.00 ± 3.25	15.57 ± 4.78	10.013	0.000 ***
	NLG	24.14 ± 2.93	19.50 ± 5.25	5.399	0.000 ***

Data are presented as mean ± standard deviation. OLG, overweight low back pain group; NLG, normal-weight low back pain group; HDL, high-density lipoproteins; LDL, low-density lipoproteins;, HOMA-IR, homeostatic model assessment-insulin resistance; CSA, cross-sectional area; ODI, Oswestry disability index. *p* < 0.05 *, *p* < 0.01 **, and *p* < 0.001 *** are statistically significant by paired *t*-test.

## Data Availability

No new data were created or analyzed in this study. Data sharing is not applicable to this article.

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
