# Peer review of "Effects of Rehabilitation Exercise on Cardiovascular Risk Factors and Muscle Cross-Sectional Area in Overweight Patients with Low Back Pain"

_healthcare, 2021, doi:10.3390/healthcare9070809_

Round 1

Reviewer 1 Report

The authors demonstrated the benefical effects of exercise in overweight indivdiuals with BMI>25 and low back pain comparing to indviduals with normal weight and low back pain. These findings are encouraging for adopting exercise intervention to improve health of the overweight individuals with chronic back pain. However, a few major comments on study design are provided below for authors’ consideration:

  1. Defintion of “obese” – you have used a BMI of 25 as the cutoff to define the obese group. However, a BMI of 25-29.9 is considered overweight rather obese (BMI>30). How many of these individuals (N=14) being studied have a BMI>30? For clarity, please include BMI in Table 2.

These current observations in individuals with BMI>25 can not be simply generalised to obese inviduals. The message being convened here is overstated. Please modify the title and texts (including methods, results and discussion) to reflect the fact of overweight rather obese inviduals were studied.

Despite no signficant differences found in BMI between the overweight and normal weight groups, the benefical effects of exercise in the former group with statistical significance were clear. I’m somehow skeptic about these findings. Can you provide evidence from other studies of comparable design to further support your findings?  

  1. Please explain further why the current exercise intervention was chosen in terms of duration (12 weeks) and intensity. In other words, why would this intervention be considered optimal and beneficial for this group of patients (BMI>25 and back pain), supported by literature?
  2. When reporting the findings, please also describe the actual concentrations or units of the parameters being measured in addition to the F statistics and P values.

Other specific comments:

  1. Please proofread and correct typos. E.g. line 20 “thrice”.
  2. Line 26, I do not understand what “period” refers to. Please use terms the scientific community would be accustomed to.
  3. Line 146, it was reported that no significant differences were found in BMI between groups. Nevertheless, a p<.001 was reported here.  

Reviewer 2 Report

In this study, Dr. Kim et al. investigated the effects of 12-week training program in a small sample of obese and normal-weight, middle-aged. women with low back pain. The design is theoretically driven by an interesting idea however some major caveat limited its impact and need to be adequately addressed.

Please provide a point-by-point response to the following main concerns:

1/ Although admitted in the limitations of the study, sample sizes look too small for inferring whatever claim. Besides, no details are reported as to the calculation (and software used) performed to determine the sample dimension.This problem is aggravated by the absence of an a-priori discussion of expected effect sizes, and corresponding power analyses. Overall, The statistical section might be definitively improved in order to properly magnify results and their interpretation.

2/ The lack of a control situation (no information of energy intake) is a general weakness. Subjects must be stringently checked in terms overall metabolism (energy expenditure, dietary intake) and possible drugs taken.
- Were participants ever asked to report eventual changes in diet and/or lifestyle? More details on energy balance and physical fitness of the subjects are required regarding their physical activity index (or calorie counts), for istance, in order to refine more homogeneously the sample. Moreover, validated questionnaires should have been used to obtain a more accurate cross-sectional snapshot for this population.

3/ I am not 100% positive about the rationale of the study. Did the Authors want to test the effects of a rehabilitative exercise-program to target both adiposity and low back pain. Is it possible to orchestrate a plan aiming at two aim points, distinctly? The introduction should logically convey the rationale of the study to the readership. Please make it solid and consistent.

4/ In the title, "muscle function" is elusive, vague and does not correspond to an outcome variable actually testes (perhaps related to physical fitness), which could have been useful, instead. Please may the Authors want to clarify and amend the title, accordingly.

I offered these critiques in a constructive spirit hoping that the Authors will find them helpful.

I found this article of potential interest for the readership of "Healthcare", however some concerns need to be fully addressed prior to considering the manuscript as suitable for publication. Specifically, I believe that the small sample size may arise doubts on the scientific soundness of the design.

Round 2

Reviewer 1 Report

I would like to thank the authors for their efforts to addressing the comments. A few more below will require your further attention.

  1. You have done a power analysis on the sample size. Could you elaborate on how the effect size of 0.27 has been estimated? Is 0.80% power a typo? Do you actually refer to 80% power which is often used in practice? However, if this is true, I do not feel these parameters would support a total of 30 samples used in the current study to reaching 80% power.
  2. I still find it hard to capture the meaning of "period" used throughout the manuscript. Are you referring to the pre- and post-intervention phase? Please correct in standard English. 
  3. The significant finding of HOMA-IR in figure 2.e has not been clearly indicated/labelled. 

Reviewer 2 Report

I thank the Authors for having attempted to address most of my raised concerns.

Author Response

The author sincerely thank you for your effort for improving quality of our manuscript.